# Designs of Charge-Balanced Edge Termination Structures for 3.3 kV SiC Power Devices Using PN Multi-Epitaxial Layers

**DOI:** 10.3390/mi16010047

**Published:** 2024-12-30

**Authors:** Sangyeob Kim, Ogyun Seok

**Affiliations:** 1Department of Semiconductor System Engineering, Kumoh National Institute of Technology, Gumi 39177, Republic of Korea; syk@kumoh.ac.kr; 2School of Electrical and Electronic Engineering, Pusan National University, Busan 46241, Republic of Korea

**Keywords:** silicon carbide (SiC), edge termination, junction termination extension (JTE), P-type epitaxial growth, trenched ring-assisted-JTE (TRA-JTE), PiN diode

## Abstract

We demonstrated 3.3 kV silicon carbide (SiC) PiN diodes using a trenched ring-assisted junction termination extension (TRA-JTE) with PN multi-epitaxial layers. Multiple P^+^ rings and width-modulated multiple trenches were utilized to alleviate electric-field crowding at the edges of the junction to quantitively control the effective charge (Q_eff_) in the termination structures. The TRA-JTE forms with the identical P-type epitaxial layer, which enables high-efficiency hole injection and conductivity modulation. The effects of major design parameters for the TRA-JTE, such as the number of trenches (N_trench_) and depth of trenches (D_trench_), were analyzed to obtain reliable blocking capabilities. Furthermore, the single-zone-JTE (SZ-JTE), ring-assisted-JTE (RA-JTE), and trenched-JTE (T-JTE) were also evaluated for comparative analysis. Our results show that the TRA-JTE exhibited the highest breakdown voltage (BV), exceeding 4.2 kV, and the strongest tolerance against variance in doping concentration for the JTE (N_JTE_) compared to both the RA-JTE and T-JTE due to the charge-balanced edge termination by multiple P^+^ rings and trench structures.

## 1. Introduction

Silicon carbide (SiC) has a wide energy bandgap and high critical electric field, so it offers significant advantages as a semiconductor material for power devices in high-voltage applications, such as high-voltage direct-current transmission systems and electric transportation [1,2,3]. SiC-based high-voltage power devices exhibit lower on-state static and switching power losses compared to the conventional Si-based thyristor due to its thinner drift layer [4].

Bipolar devices are suitable for high-voltage applications due to the availability of conductivity modulation, provided that sufficient hole injection occurs from the anode to the drift layer [5,6,7]. However, the Al implantation process, typically employed for the formation of P-type anodes, is known to introduce various lifetime degrading defects, including Z_1/2_ and basal plane dislocation (BPD) [8,9,10,11,12,13,14,15]. These defects can lead to bipolar degradation, significantly affecting both the performance and long-term reliability of bipolar devices [10,11].

Appropriate edge termination designs are required to mitigate electric-field crowding at the edges of the PN junction. Field-limiting rings (FLRs) and a junction termination extension (JTE) are commonly used structures for SiC power devices [16,17,18,19,20,21,22,23,24,25,26,27,28,29,30]. In the case of FLRs, multiple P^+^ rings are formed simultaneously with the main junction. However, FLRs require a large chip area to sustain high voltage. On the other hand, a JTE can be implemented with a smaller chip area than FLRs by achieving charge balance between the JTE and drift layers. A JTE is required to ensure high breakdown voltage (BV) over a wide range of JTE concentration (N_JTE_) and fixed oxide charge (Q_F_).

Several research groups have reported high-voltage SiC PiN diodes incorporating PN multi-epitaxial layers to enhance conductivity modulation and prevent the devices from degradation-related problems [31,32,33,34,35,36,37]. Nevertheless, in the reported studies, the SiC PiN devices employing PN multi-epitaxial layers typically utilized mesa etching to form conventional edge termination structures on the drift layers, such as FLRs and JTEs, using Al implantation [16,17,18,19,20,21,22,23,24,25,26,27,28,29,30]. The use of mesa etching in P-type epitaxial layers to form edge termination structures complicates dimension reduction and induces relatively complex processes compared to the planar-type structures [31,32,33,34,35,36,37].

In this paper, we propose 3.3 kV SiC PiN diodes featuring a trenched ring-assisted junction termination extension (TRA-JTE) using PN multi-epitaxial layers. The TRA-JTE design benefits from the planar nature of P-type epitaxial growth, which facilitates easier design and is expected to enhance hole injection. We evaluated the BVs and electric field distributions of the TRA-JTE structure and compared them with those of the single-zone-JTE (SZ-JTE), ring-assisted-JTE (RA-JTE), and trenched-JTE (T-JTE). The TRA-JTE structure incorporates multiple P^+^ rings and trench-etched regions to effectively control the effective charge (Q_eff_). We successfully demonstrated the edge termination structure using PN multi-epitaxial layers for high-voltage mesa-shaped devices.

## 2. Designs and Structures

Process simulation to design various edge termination structures with PN multi-epitaxial layers and analyze their electrical characteristics was carried out using Sentaurus TCAD (T-2022.03). Figure 1 shows the cross-sectional views of 3.3 kV SiC PiN diodes with (a) SZ-JTE, (b) RA-JTE, (c) T-JTE, and (d) TRA-JTE. The sequence for process simulation of devices was followed. Initially, a 30 μm thick N-type drift layer with a doping concentration of 3 × 10^15^ cm^−3^ was epitaxially grown on an N^+^ substrate, followed by the growth of a 2 μm thick P-type epitaxial layer on the N-type drift layer for use of both hole-injection active regions and JTE structures in edge termination regions. After that, P^+^ regions formed high-dose Al implantation, simultaneously constructing multiple P^+^ rings with a width of 3 μm for both RA-JTE and TRA-JTE.

The parameter S_1_, defined as the spacing between the edge of the main junction and the first P^+^ ring, was optimized to 3 μm, while the increment in spacing between subsequent P^+^ rings, denoted S_i_, was set to 0.5 μm. The width of the JTE (W_JTE_) was determined by the mesa etching process and fixed at 135 μm. For the SZ-JTE and RA-JTE structures, a 2.1 μm deep etching of the SiC was performed, completely removing the P-type epitaxial layers in the outer JTE regions, accounting for a 5% over-etching margin. In case of T-JTE and TRA-JTE, a 1.5 μm deep trench etching was carried out prior to the subsequent 0.6 μm deep mesa etching, as illustrated in Figure 2. The optimized design parameters for each structure are summarized in Table 1.

## 3. Results and Discussion

The BVs of the SZ-JTE and RA-JTE structures as a function of N_JTE_ are compared in Figure 3. The RA-JTE exhibits greater tolerance to variations in N_JTE_ compared to the SZ-JTE. While the SZ-JTE has the benefit of being simply formed through a single mesa etching process, it exhibits high sensitivity to N_JTE_. As a result, the BVs of the SZ-JTE structure remain below 3.3 kV, except when the N_JTE_ value is optimized to 5 × 10^16^ cm^−3^. In the RA-JTE structure, the presence of multiple P^+^ rings enhances the electric field distribution near the main junction, particularly at low N_JTE_ values.

Figure 4 shows the electric field distributions of the SZ-JTE and RA-JTE at a reverse bias of 3.0 kV for the N_JTE_ value of 4 × 10^16^ cm^−3^. The SZ-JTE shows a high electric field value at the main junction, whereas the P^+^ rings of the RA-JTE effectively suppress the electric-field crowding at the main junction.

At a high N_JTE_ exceeding 5 × 10^16^ cm^−3^, both the SZ-JTE and RA-JTE exhibit sharp reductions in BV due to the significant electric-field crowding at the mesa sidewall, as illustrated in Figure 5. When the electric field concentrates at the mesa edge, avalanche breakdown may occur at undesirably low voltage or irreversible physical damage to the field oxide layer. In the proposed TRA-JTE structure, multiple trenches were additionally employed to optimize the electric field distribution, specifically mitigating crowding at the mesa sidewall to ensure stable blocking performance.

Figure 6 shows the BVs of the SZ-JTE, RA-JTE, T-JTE, and TRA-JTE according to N_JTE_. Both the T-JTE and TRA-JTE have high BVs exceeding the rated 3.3 kV up to an N_JTE_ of 1.7 × 10^17^ cm^−3^, attributed to the effective charge balancing achieved by the trench structure in the P-type epitaxial layers. Our simulation results exhibit that the electric field is effectively distributed among three distinct peaks located at the P^+^ rings, double trenches, and mesa sidewall.

Effective charge density (Q_eff_ = N_JTE_ × T_P-epi_) was used to analyze the charge balance in the edge termination structures, including the SZ-JTE, RA-JTE, T-JTE, and TRA-JTE. We classified the specific regions within the edge termination structures based on the Q_eff_ as illustrated in Figure 7.

The SZ-JTE exhibits a single region of medium Q_eff_ with a uniform charge distribution, resulting in highly sensitive breakdown characteristics to variation in N_JTE_. The RA-JTE and TRA-JTE have high Q_eff_ regions attributed to the presence of P^+^ rings so that the potential of the main junction is effectively delivered outward through the termination structure, resulting in stable blocking characteristics even at low N_JTE_. The T-JTE and TRA-JTE have low Q_eff_ regions due to the locally thinned P-type epitaxial layer. The trenched regions create multiple electric field peaks to support blocking voltage and a gradual increase in potential. In the case of the TRA-JTE, the presence of all the types of Q_eff_ regions offers advantages for achieving charge balance and brings about robust blocking characteristics against variations in N_JTE_.

Figure 8a illustrates the electric field distributions for the TRA-JTE at a reverse bias of 3.3 kV with various values of N_JTE_. Regions 1, 2, and 3 include the main junction/multiple P^+^ rings, double trenches/the first mesa, and second mesa, respectively. The multiple P^+^ rings effectively suppress the electric-field crowding near the main junction in the case of an N_JTE_ of 5 × 10^16^ cm^−3^. For higher N_JTE_ conditions than of 5 × 10^16^ cm^−3^, the double trenches significantly alleviate the electric field at the mesa sidewall through the charge-balanced P-type epitaxial layer. Figure 8b–e illustrate the locations of the maximum electric field observed in each region of the TRA-JTE structure under various N_JTE_ conditions with the reverse bias of 3.3 kV. As N_JTE_ increases from 5 × 10^16^ to 1.7 × 10^17^ cm^−3^, the maximum electric field in region 1 decreases from 2.10 to 1.90 MV/cm, while that corresponding value in region 3 increases from 0.58 MV/cm to 2.53 MV/cm. It is noted that the TRA-JTE structures demonstrate reliable blocking capability under relatively high N_JTE_ conditions by controlling the charge balance in region 2 through the double trenches so that the electric field peak values are almost evenly distributed across the entire edge termination structure under a higher N_JTE_ of 5 × 10^16^ cm^−3^.

Figure 9 shows the electric field distributions of the SZ-JTE, RA-JTE, T-JTE, and TRA-JTE at breakdown for the N_JTE_ value of 5 × 10^16^ cm^−3^ with BVs above 3.3 kV for all the structures. In the case of the SZ-JTE and RA-JTE, the electric field is concentrated at the mesa sidewall. In contrast, for the T-JTE and TRA-JTE, the electric field at the mesa is effectively suppressed because the JTE region is fully depleted prior to the occurrence of avalanche breakdown. As a result, the T-JTE and TRA-JTE exhibit peak values of electric field of less than 2 MV/cm, lower than those of the SZ-JTE and RA-JTE. Notably, the TRA-JTE has the widest N_JTE_ tolerance, surpassing the T-JTE by 8.33%, attributed to the high Q_eff_ at the multiple P^+^ rings.

Figure 10 shows the forward J–V characteristics of the SZ-JTE, RA-JTE, T-JTE, and TRA-JTE for the N_JTE_ value of 5 × 10^16^ cm^−3^. Since the main junction is identical for all the structures, the forward voltage drops at a current density of 100 A/cm^2^ are 3.3 V for all the structures. These results demonstrate that the TRA-JTE exhibits the most outstanding reverse characteristics without sacrificing forward characteristics.

Q_F_ is formed in field oxide during thermal oxidation and post-oxidation annealing [38,39,40]. It affects the Q_eff_, potentially causing a change in the BVs of SiC devices [41,42,43]. Figure 11 shows the simulation results for the SZ-JTE, RA-JTE, T-JTE, and TRA-JTE structures considering several Q_F_ values in the field oxide. Positive Q_F_ values attract electrons to the surface, reducing Q_eff_ and decreasing the depletion curvature at the outer edge of the JTE. The incorporation of multiple P^+^ rings in the RA-JTE and TRA-JTE structures effectively compensates for the reduced Q_eff_. As a result, in the presence of positive Q_F_, the BVs of the RA-JTE and TRA-JTE are reduced by only 0.30 kV and 0.66 kV, respectively, whereas the BVs of the SZ-JTE and T-JTE are reduced by 1.07 kV and 1.01 kV, respectively.

A negative Q_F_ value increases the Q_eff_, causing the depletion regions to extend towards the end of the JTE. Consequently, the BVs of the SZ-JTE and RA-JTE are sharply reduced under the condition of a negative Q_F_. However, the T-JTE and TRA-JTE exhibit reduced sensitivity to negative Q_F_ due to their thinner T_P-epi_ by the formation of trenches, which maintain a lower Q_eff_ compared to other structures. It can be observed that the BV of the T-JTE is increased from 4.1 kV to 4.3 kV. Also, the stability of the TRA-JTE was validated, demonstrating relatively less sensitivity to both positive and negative Q_F_.

Subsequently, we simulated the TRA-JTE under various design parameters, including the number of trenches (N_trench_) and depth of trenches (D_trench_). Figure 12 illustrates the BVs of the TRA-JTE according to N_JTE_ and N_trench_ with identical D_trench_ of 1.5 µm with a comparison of the results from the RA-JTE. The BVs of the RA-JTE structure are sharply reduced at N_JTE_ of 6 × 10^16^ cm^−3^ because the electric field is crowded at the mesa sidewall, as shown in Figure 5. In the case of the TRA-JTE with a single trench of 3 µm wide width, the BVs are slightly higher compared to the RA-JTE. However, the TRA-JTE with a single trench is insufficient to effectively distribute the electric field for various N_JTE_ conditions. Our simulation results indicate that the TRA-JTE with double trenches demonstrates the widest N_JTE_ tolerance range even though the structure with triple trenches exhibits slightly higher BV values at N_JTE_ below 1.6 × 10^17^ cm^−3^.

Figure 13 shows the BVs as a function of N_JTE_ for the TRA-JTE with D_trench_ values of 0.5 µm, 1.0 µm, 1.5 µm, and 2.0 µm. In all the cases, the double trenches are employed. The device with D_trench_ of 0.5 µm has slightly higher BVs at low N_JTE_, attributed to the relatively high Q_eff_ in the trenched region compared to the other D_trench_ conditions. However, the BVs are steeply decreased when N_JTE_ is higher than 7 × 10^16^ cm^−3^. In the N_JTE_ range between 8 × 10^16^ cm^−3^ and 1.0 × 10^17^ cm^−3^, the TRA-JTE with D_trench_ of 1.0 µm shows the highest BVs and N_JTE_ range between 1.1 × 10^16^ cm^−3^ and 1.7 × 10^17^ cm^−3^, and the TRA-JTE with D_trench_ of 1.5 µm shows the highest BVs. The larger D_trench_ enhances the spreading depletion in a full JTE region, including a mesa sidewall. However, at D_trench_ of 2.0 µm, too much potential is delivered from the etched region as the P-type epitaxial layer is almost removed.

Figure 14 shows the electric field distributions of the TRA-JTE with D_trench_ values of (a) 0.5 µm, (b) 1.0 µm, (c) 1.5 µm, and (d) 2.0 µm at the N_JTE_ value of 7 × 10^16^ cm^−3^. As D_trench_ increases from 0.5 µm to 2.0 µm, the peak electric field points shift from the second mesa to the first trench. Notably, the PiN diode with D_trench_ of 1.5 µm exhibits the widest N_JTE_ tolerance. These results indicate that D_trench_ is a critical design parameter for achieving the charge balance of the TRA-JTE. Figure 13 shows the electric field distributions of the TRA-JTE with D_trench_ values of (a) 0.5 µm, (b) 1.0 µm, (c) 1.5 µm, and (d) 2.0 µm at the N_JTE_ value of 7 × 10^16^ cm^−3^. As D_trench_ increases from 0.5 µm to 2.0 µm, the peak electric field points shift from the second mesa to the first trench. Notably, the PiN diode with D_trench_ of 1.5 µm exhibits the widest N_JTE_ tolerance. Those results indicate that D_trench_ is a critical design parameter for achieving the charge balance of the TRA-JTE.

## 4. Conclusions

We designed the SZ-JTE, RA-JTE, T-JTE, and TRA-JTE using PN multi-epitaxial layers as JTEs for 3.3 kV SiC PiN diodes. The RA-JTE and TRA-JTE exhibited high BVs at low N_JTE_ due to the multiple P^+^ rings enhancing the Q_eff_ near the main junction. Additional trench structures for the T-JTE and TRA-JTE effectively sustained high BVs at high N_JTE_ and are less sensitive to Q_F_ variation without increasing the forward voltage drop. By optimizing the parameters of the TRA-JTE, including N_trench_ and D_trench_, we demonstrated that the TRA-JTE with double trenches and D_trench_ of 1.5 µm is a suitable edge termination structure for high-voltage SiC devices.

## Figures and Tables

**Figure 1 micromachines-16-00047-f001:**
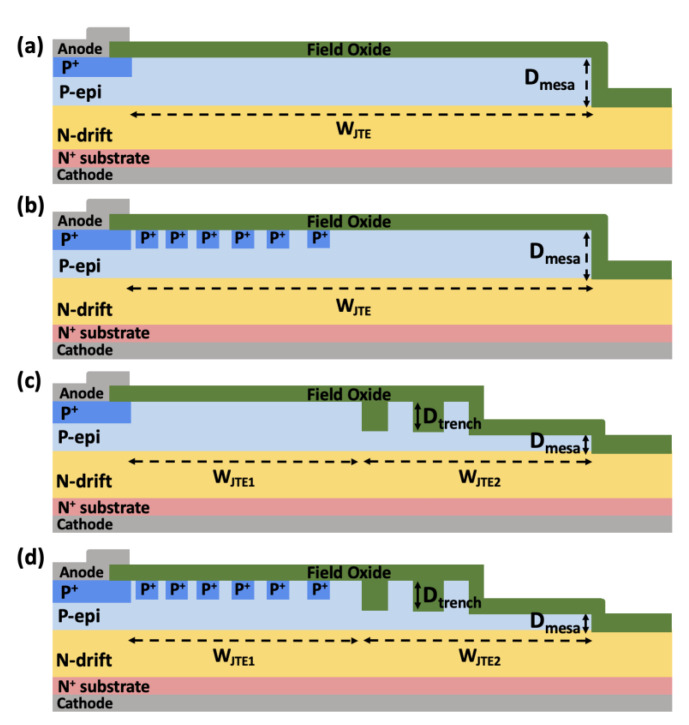
Cross-sectional views of (**a**) SZ-JTE, (**b**) RA-JTE, (**c**) T-JTE, and (**d**) TRA-JTE.

**Figure 2 micromachines-16-00047-f002:**
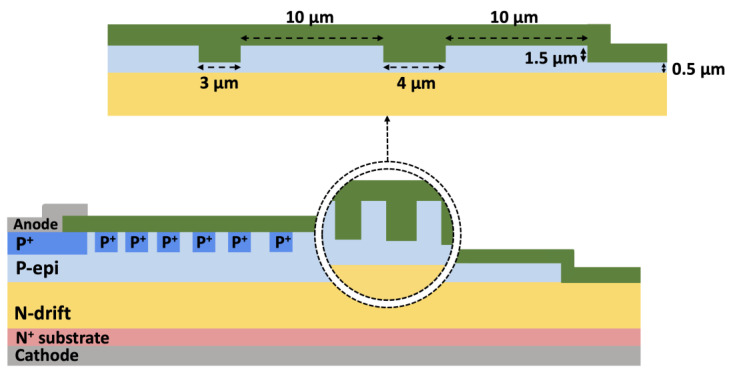
Widths and spaces between double trenches of TRA-JTE.

**Figure 3 micromachines-16-00047-f003:**
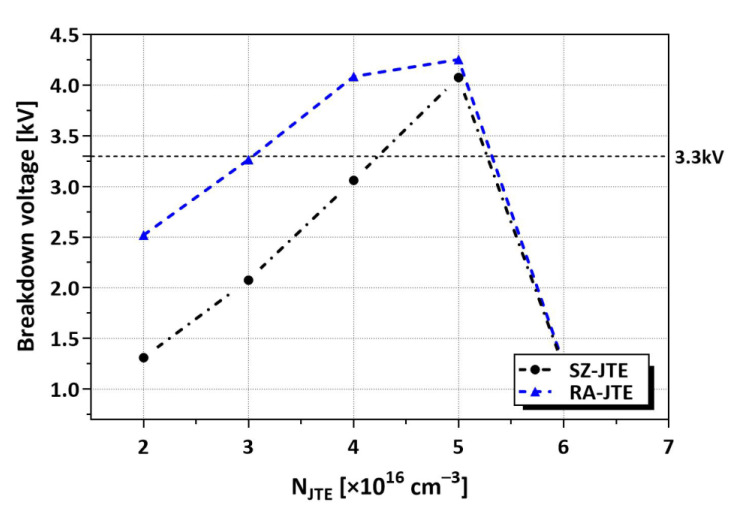
BVs according to N_JTE_ for SZ-JTE and RA-JTE. N_JTE_ values are increased by 1 × 10^16^ cm^−3^ from 2 × 10^16^ cm^−3^ to 6 × 10^16^ cm^−3^.

**Figure 4 micromachines-16-00047-f004:**
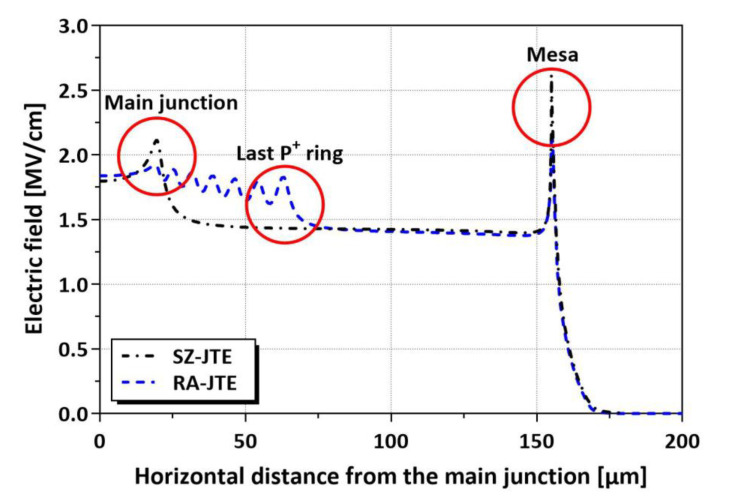
Electric field distributions of SZ-JTE and RA-JTE at reverse bias of 3.0 kV for N_JTE_ value of 4 × 10^16^ cm^−3^.

**Figure 5 micromachines-16-00047-f005:**
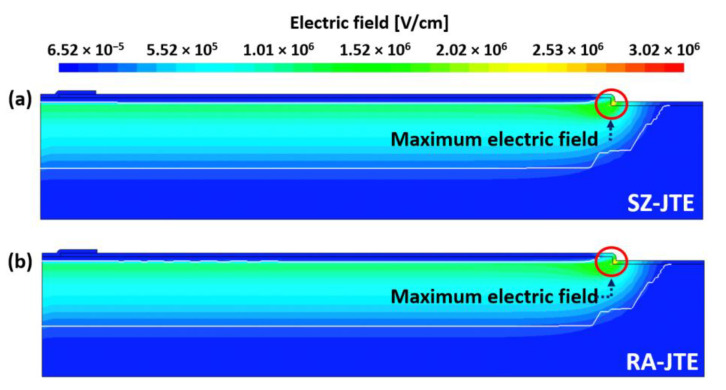
Electric field distributions of (**a**) SZ-JTE and (**b**) RA-JTE at reverse bias of 1.2 kV of N_JTE_ value of 6 × 10^16^ cm^−3^.

**Figure 6 micromachines-16-00047-f006:**
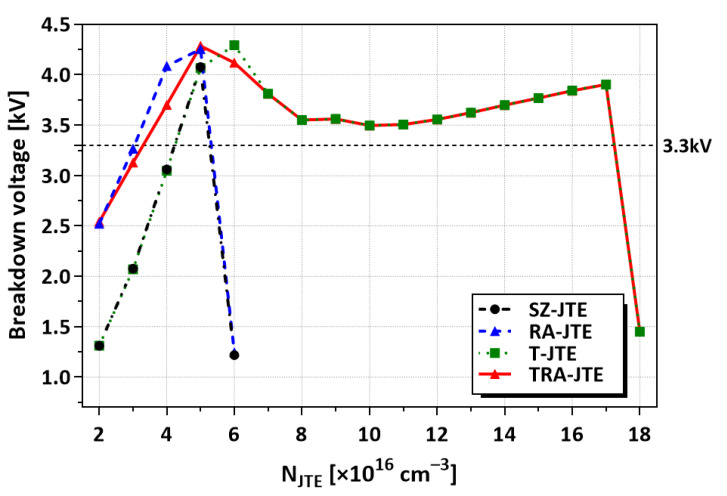
BVs according to N_JTE_ for SZ-JTE, RA-JTE, T-JTE, and TRA-JTE.

**Figure 7 micromachines-16-00047-f007:**
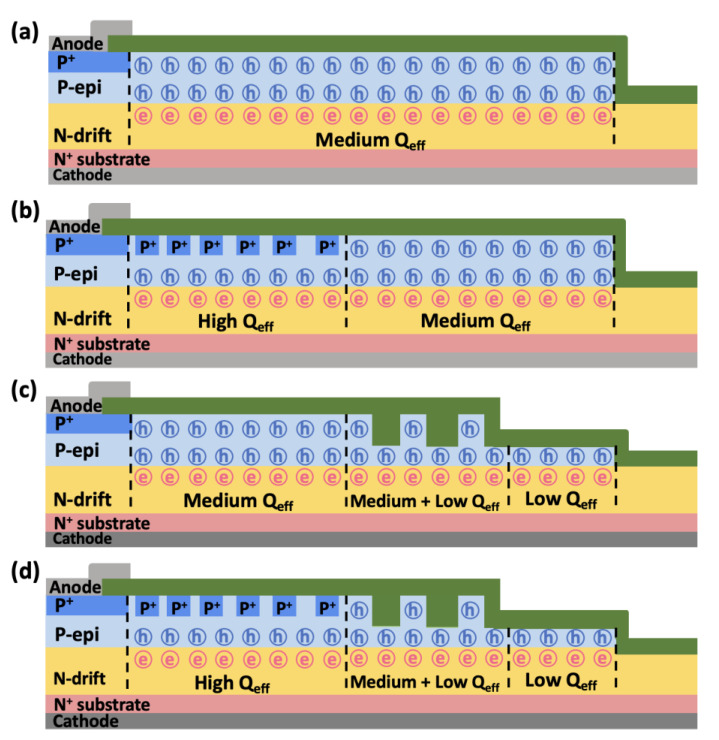
Cross-sectional views of Q_eff_ including electrons and holes per unit area for (**a**) SZ-JTE, (**b**) RA-JTE, (**c**) T-JTE, and (**d**) TRA-JTE.

**Figure 8 micromachines-16-00047-f008:**
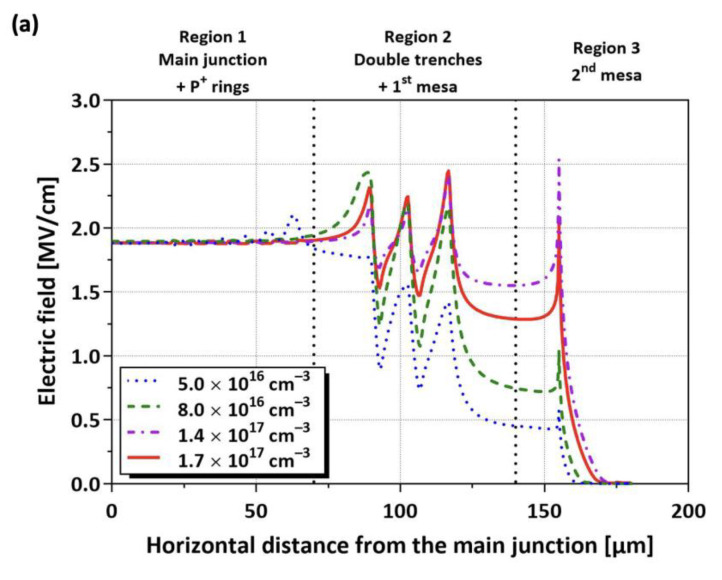
(**a**) Electric field distributions for TRA-JTE at reverse bias of 3.3 kV with various N_JTE_ and peak electric field of TRA-JTE per divided region for N_JTE_ values of (**b**) 5 × 10^16^ cm^−3^, (**c**) 8 × 10^16^ cm^−3^, (**d**) 1.4 × 10^17^ cm^−3^, and (**e**) 1.7 × 10^17^ cm^−3^.

**Figure 9 micromachines-16-00047-f009:**
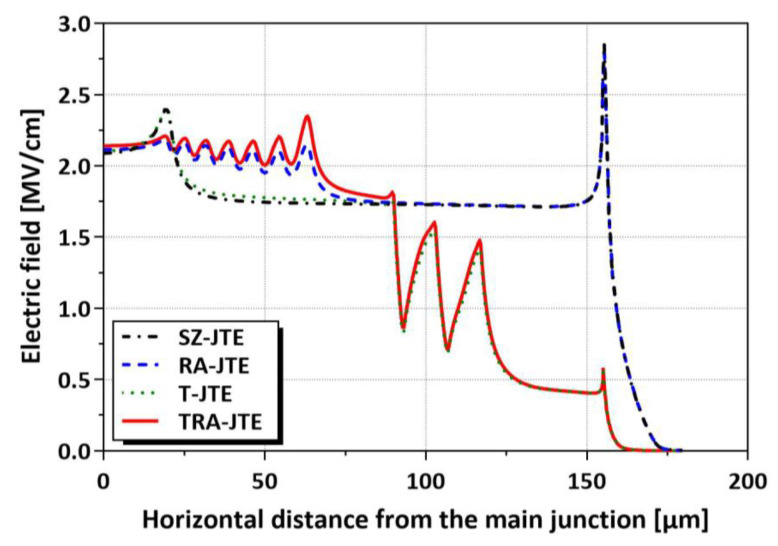
Electric field distributions of SZ-JTE, RA-JTE, T-JTE, and TRA-JTE at breakdown for N_JTE_ value of 5 × 10^16^ cm^−3^.

**Figure 10 micromachines-16-00047-f010:**
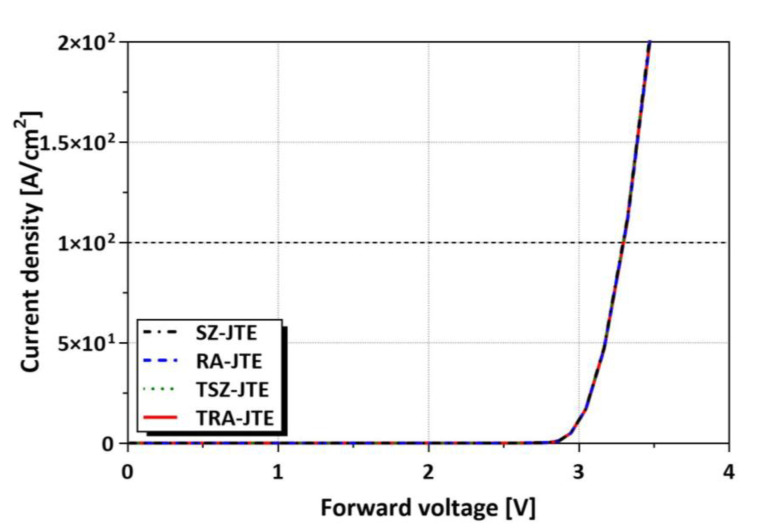
J–V characteristics of the SZ-JTE, RA-JTE, T-JTE, and TRA-JTE for the N_JTE_ value of 5 × 10^16^ cm^−3^.

**Figure 11 micromachines-16-00047-f011:**
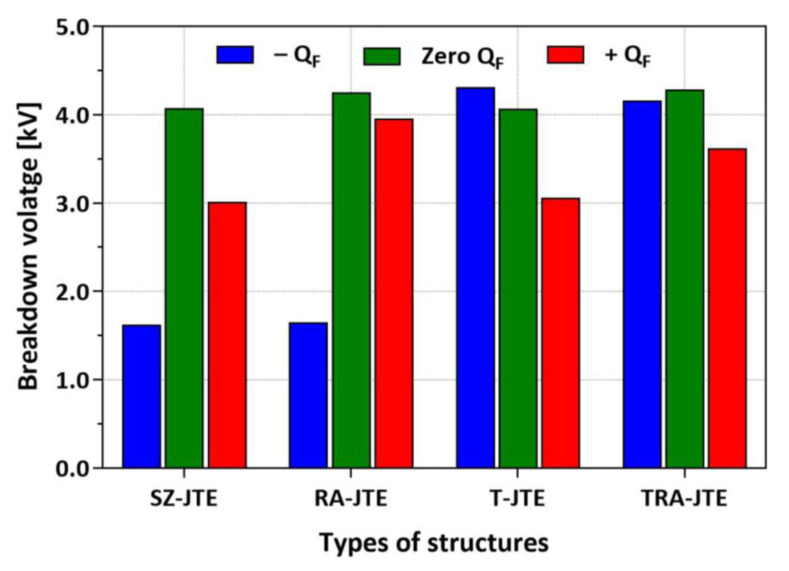
BVs according to Q_F_ of ±2 × 10^12^ cm^−2^ for SZ-JTE, RA-JTE, T-JTE, and TRA-JTE at N_JTE_ value of 5 × 10^16^ cm^−3^.

**Figure 12 micromachines-16-00047-f012:**
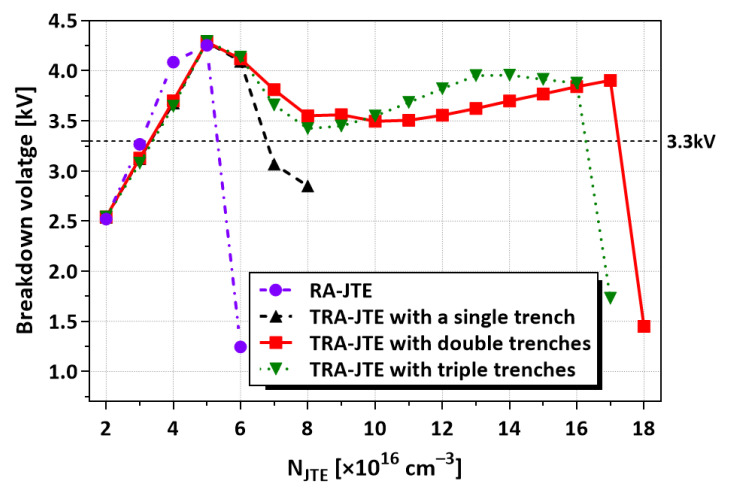
BVs according to N_JTE_ and N_trench_ for RA-JTE and TRA-JTE with identical D_trench_ of 1.5 µm.

**Figure 13 micromachines-16-00047-f013:**
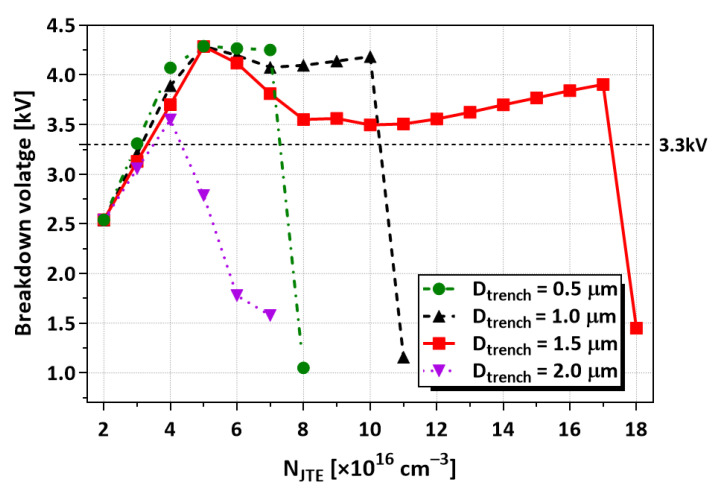
BVs according to N_JTE_ and D_trench_ for TRA-JTE with double trenches.

**Figure 14 micromachines-16-00047-f014:**
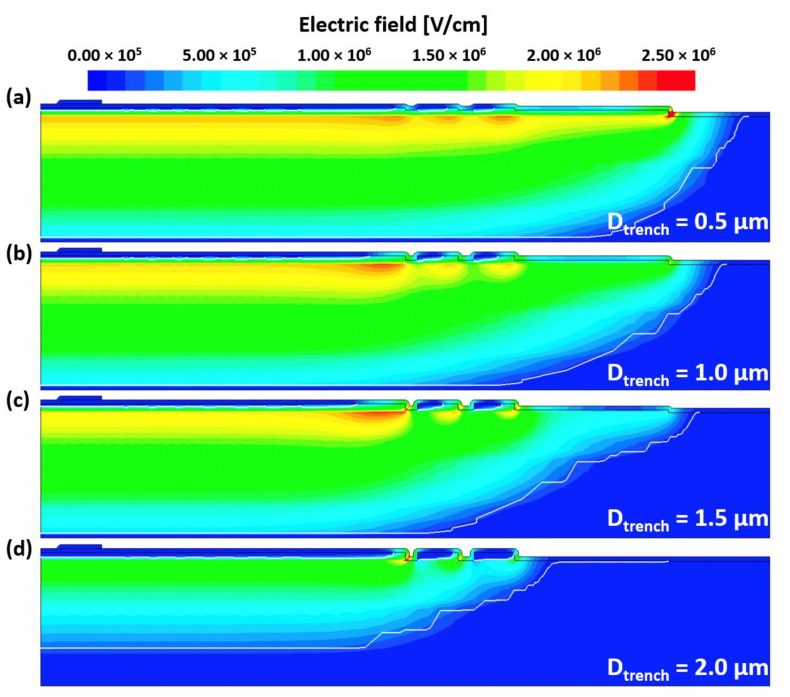
Electric field distributions of TRA-JTE with D_trench_ values of (**a**) 0.5 µm, (**b**) 1.0 µm, (**c**) 1.5 µm, and (**d**) 2.0 µm at N_JTE_ value of 7 × 10^16^ cm^−3^.

**Table 1 micromachines-16-00047-t001:** Optimized design parameters of each structure.

Structures	W_JTE_ [μm]	D_mesa_ [μm]	D_trench_ [μm]	W_ring_ [μm]	S_1_ [μm]	S_i_ [μm]
SZ-JTE	135	2.1	-	-	-	-
RA-JTE	135	2.1	-	3	3	0.5
T-JTE	W_JTE1_ = 70	0.6	1.5	-	-	-
	W_JTE2_ = 65					
TRA-JTE	W_JTE1_ = 70	0.6	1.5	3	3	0.5
	W_JTE2_ = 65					

## Data Availability

All data are presented in this paper in the form of figures.

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
