# Peer review of "Designs of Charge-Balanced Edge Termination Structures for 3.3 kV SiC Power Devices Using PN Multi-Epitaxial Layers"

_micromachines, 2024, doi:10.3390/mi16010047_

Round 1

Reviewer 1 Report

Comments and Suggestions for Authors

The authors demonstrated 3.3 kV Silicon Carbide (SiC) PiN diodes using a trenched ring assisted junction termination extension (TRA-JTE) with PN multi-epitaxial layers. results show that TRA-JTE exhibited the highest breakdown voltage (BV) exceeding 4.2 kV and the strongest tolerance against variance of doping concentration for JTE (NJTE) compared to both RA-JTE and T-JTE due to the charge balanced edge termination by multiple P+ rings and trench structures. The study is very interesting. The research is presented clearly, the paper could be published.

Author Response

Comments 1: The authors demonstrated 3.3 kV Silicon Carbide (SiC) PiN diodes using a trenched ring assisted junction termination extension (TRA-JTE) with PN multi-epitaxial layers. results show that TRA-JTE exhibited the highest breakdown voltage (BV) exceeding 4.2 kV and the strongest tolerance against variance of doping concentration for JTE (NJTE) compared to both RA-JTE and T-JTE due to the charge balanced edge termination by multiple P+ rings and trench structures. The study is very interesting. The research is presented clearly, the paper could be published.

Response 1: Thank you for your positive feedback on our manuscript. We greatly appreciate your recognition of our work on the development of 3.3 kV SiC PiN diodes with the trenched ring assisted junction termination extension (TRA-JTE). We are delighted to hear that you find the study interesting and the manuscript suitable for publication.

Reviewer 2 Report

Comments and Suggestions for Authors

In this paper, the authors have proposed a PiN diode design based on SiC using a trenched ring assisted junction termination extension (TRA-JTE) with PN multi-epitaxial layers. The main idea is to use P+ regions and width-modulated multiple trenches, which noticeably improve the properties of the PiN diode. The construction proposed enables high-efficiency hole injection and conductivity modulation. In principle, the main idea is presented quite simply and clearly, the simulated characteristics are presented in detail in the figures, where the advantages of the proposed design of the PIN diode can be seen. It is concluded that this design increases the breakdown voltage up to 4.2 kV. I believe that this work will be of interest to the readers of the journal, so the article can be published.

Author Response

Comments 1: In this paper, the authors have proposed a PiN diode design based on SiC using a trenched ring assisted junction termination extension (TRA-JTE) with PN multi-epitaxial layers. The main idea is to use P+ regions and width-modulated multiple trenches, which noticeably improve the properties of the PiN diode. The construction proposed enables high-efficiency hole injection and conductivity modulation. In principle, the main idea is presented quite simply and clearly, the simulated characteristics are presented in detail in the figures, where the advantages of the proposed design of the PIN diode can be seen. It is concluded that this design increases the breakdown voltage up to 4.2 kV. I believe that this work will be of interest to the readers of the journal, so the article can be published.

Response 1: Thank you for your positive feedback on our manuscript. We greatly appreciate your recognition of our work on the development of 3.3 kV SiC PiN diodes with the trenched ring assisted junction termination extension (TRA-JTE). We are delighted to hear that you find the study interesting and the manuscript suitable for publication.

Reviewer 3 Report

Comments and Suggestions for Authors

The authors proposed a trenched ring assisted junction termination extension (TRA-JTE) to demonstrate 3.3 kV Silicon Carbide (SiC) PiN diodes, which is interesting. The following questions should be addressed.

1. Will TRA-JTE need more chip area compared with the conventional termination structures using field limiting ring (Solid State Electronics 2020, 171, 107873)? The authors should make a comment.

2. How about the on-resistant and turn-on voltage of the proposed PiN diodes? Will TRA-JTE affect on-resistant or turn-on voltage?  

Round 2

Reviewer 3 Report

Comments and Suggestions for Authors

In my previous review, I wonder whether TRA-JTE need more chip area compared with the conventional termination structures using field limiting rings (FLRs) and the authors should make a comment. In the revised manuscript, the added references are basically focused on JTE rather than FLRs although some references used FLRs to assist JTE. For the fairness of the statement, the author should add the comparison with the reported termination structures using FLRs (Solid State Electronics 2020, 171, 107873). 
